# Factors Affecting Antibiotic Prescription among Hospital Physicians in a Low-Antimicrobial-Resistance Country: A Qualitative Study

**DOI:** 10.3390/antibiotics11010098

**Published:** 2022-01-13

**Authors:** Ingrid Christensen, Jon Birger Haug, Dag Berild, Jørgen Vildershøj Bjørnholt, Brita Skodvin, Lars-Petter Jelsness-Jørgensen

**Affiliations:** 1Department of Infection Control, Østfold Hospital Trust, Kalnes, 1714 Graalum, Norway; jonhau@so-hf.no (J.B.H.); lars.p.jelsness-jorgensen@hiof.no (L.-P.J.-J.); 2Faculty of Medicine, Ph.D. Program Medicine and Health Sciences, University of Oslo, 0315 Oslo, Norway; 3Department of Infectious Diseases, Oslo University Hospital, 0424 Oslo, Norway; dag.berild@medisin.uio.no; 4Faculty of Medicine, Institute of Clinical Medicine, University of Oslo, 0315 Oslo, Norway; j.v.bjornholt@medisin.uio.no; 5Department of Microbiology, Oslo University Hospital, 0424 Oslo, Norway; 6Norwegian Advisory Unit on Antibiotic Use in Hospitals, Department of Research and Development, Haukeland University Hospital, 5021 Bergen, Norway; brita.skodvin@helse-bergen.no; 7Faculty of Health and Social Studies, Østfold University College, 1671 Fredrikstad, Norway

**Keywords:** hospital physicians, low-resistance country, antibiotic prescription, antibiotic stewardship, antimicrobial resistance, semi-structured interviews, qualitative study

## Abstract

Antimicrobial resistance (AMR) is a threat to hospital patients. Antimicrobial stewardship programs (ASPs) can counteract AMR. To optimize ASPs, we need to understand what affects physicians’ antibiotic prescription from several contexts. In this study, we aimed to explore the factors affecting hospital physicians’ antibiotic choices in a low-resistance country to identify potential targets for future ASPs. We interviewed 14 physicians involved in antibiotic prescription in a Norwegian hospital. The interviews were audiotaped, transcribed verbatim, and analyzed using thematic analysis. The main factors affecting antibiotic prescription were a high work pressure, insufficient staff resources, and uncertainties regarding clinical decisions. Treatment expectations from patients and next of kin, benevolence towards the patients, suboptimal microbiological testing, and limited time for infectious disease specialists to offer advisory services also affected the antibiotic choices. Future ASP efforts should evaluate the system organization and prioritizations to address and manage potential time-pressure issues. To limit the use of broad-spectrum antibiotics, improving microbiology testing and the routines for consultations with infectious disease specialists seems beneficial. We also identified a need among the prescribing physicians for a debate on ethical antibiotic questions.

## 1. Introduction

Antibiotics are a cornerstone in the treatment of infections and critical to modern medicine [1]. However, inappropriate antibiotic use has led to increased antimicrobial resistance (AMR), threatening global health [2,3,4]. Antimicrobial stewardship programs (ASPs), defined as “a coherent set of actions which promote using antimicrobials responsibly” [5], are essential to combat AMR [6]. Understanding and considering the factors influencing physicians’ antibiotic decisions are critical to provide tailored measures towards the stakeholders and thus the success of ASPs [7,8]. Systematic reviews on antibiotic prescribing behavior have identified some common barriers to rational prescription, including the fear of negative consequences for the patient and the belief that other professionals or other countries and health care practices, rather than themselves, are causing the problem of AMR [9,10,11]. ASPs are context-specific and must be adjusted to different health care environments. Exploration of key drivers from several contexts are crucial to ensure robust interventions [7,8]. Consequently, there is a need for exploratory studies across various settings [12,13].

In Norway, broad-spectrum antibiotics (BSAs) and AMR rates are lower than in most other countries. For instance, Norway has the lowest rate of methicillin-resistant staphylococcus aureus and a lower rate of other resistant bacteria, in company with Denmark and Finland, compared to southern and eastern Europe [14]. The main reasons are the historical strict national regulatory approval of new antibiotics and conservative guidelines for antibiotic use [15]. However, there has been a slight but steady increase in total and broad-spectrum antibiotic use in Norwegian hospitals, on a grander scale than the prevalence of infectious diseases suggests [16]. At the hospital where this study was conducted, BSA consumption has been relatively high compared to other Norwegian hospitals, which highlighted a need for reinforced ASPs.

The study aim of the present study was to explore factors affecting hospital physicians’ antibiotic prescribing practices, in a low-resistance country, to identify potential targets for improvement of future ASPs.

## 2. Results

The final study population included 14 respondents: five physicians from infectious diseases; three physicians from oncology; one medical resident; and one physician from anesthesiology, gastrointestinal surgery, gastroenterology, pulmonology, and hematology. The participants’ experience level and characteristics are presented in Table 1.

We identified four main themes affecting the physicians’ antibiotic prescribing choices: (i) Clinical workflow pressures, (ii) Clinical uncertainty, (iii) Decision support, and (iv) Benevolence. Visualization of the themes and subthemes is shown in Figure 1. The themes, their respective subthemes, and illustrative quotes are shown in Table 2.

### 2.1. Theme (i) Clinical Workflow Pressures

The theme “Clinical workflow pressures” consists of the subthemes “time pressure” and “follow-up resources”. Both subthemes represent physicians’ experience of unfulfilled intentions of making thorough antibiotic decisions. The physicians constantly experienced being in a squeeze between providing optimal patient care, including making rational antibiotic decisions, and having a high workload and limited follow-up resources.

“Time pressure” reflects the pressures upon physicians’ to increase patient turnover. The respondents reported experiencing a pressure mainly from nursing staff due to bed pressure. The physicians also reported putting pressure on themselves, e.g., when they observed many patients waiting in the emergency room. This strain was reported to often result in unnecessary use of broad-spectrum antibiotics to compensate for short time to observe patients and to perform adequate diagnostics:


*“I think a lot is due to the pressure in the hospital; we hear all the time, ‘Discharge, discharge, discharge’ (…) We get whipped up like that, (…) and then I think the threshold (…) to give broader spectrum antibiotics is lower.”*
(Respondent 11, 22 years of experience).

“Follow-up resources” refers to the physicians’ experience of a lack of patient follow-up resources which made them scared of patient adverse outcomes and thus led to use of more antibiotics to ensure safety. A lack of nurses (and thus capacity) was, for instance, viewed as a challenge:


*“It is important that the patient is properly followed up; that is what’s scary in an understaffed hospital full of patients: a lack of nurses. Then, it might be more rational to use broader spectrum antibiotics than in a more orderly situation (…) A typical example is that we actually shift from penicillin to cefotaxime (i.e., a more broad-spectrum antibiotic) on Fridays due to lesser capacity during weekends”*
(R7, 13 years).

### 2.2. Theme (ii) Clinical Uncertainty

The theme “Clinical uncertainty” consisted of two subthemes: “just in case” and “knowledge and experience”. Both subthemes underscored that the most important aspect for the physicians when they prescribed antibiotics was to feel certain in their decisions, both regarding a favorable outcome for their patients and to justify their decisions to their peers later.

“Just in case” reflects the finding that if the physicians were uncertain whether the clinical condition would deteriorate at home, they reported they would often continue to give antibiotics, even broad-spectrum antibiotics, to be on the safer side:


*“We know that, theoretically, they often don’t need antibiotics, but it is hard to select the patients who actually don’t need antibiotics and have the courage to send them home without any (…) you are scared the patient will come back in worse shape, and you will be asked why you didn’t continue antibiotics”*
(R12, 10 years).

The physicians also considered their own “knowledge and experience” levels when making antibiotic decisions. The less experienced (under five years of experience) physicians more often reported that a feeling of inadequate knowledge led to less confidence to prescribe restrictively (please see Table 3 for illustrative quotes). They expressed that this lack of confidence led to an increased use of antibiotics just to be “on the safe side”. In contrast, having enough experience to ‘trust their gut’ facilitated a more restrictive prescription:


*“It is partly the experience level that allows you to have the guts to be restrictive”*
(R7, 13 years).

### 2.3. Theme (iii) Decision Support

“Decision support” consists of the subthemes “microbiological tests” and “collegial consulting”. These subthemes illustrated that the physicians viewed microbiological test and consultation as facilitators of rational prescription, but experienced that these facilitators were underused due to capacity constraints. Thus, they reported that underuse of available decision support negatively affected antibiotic prescription.

The physicians considered microbiological tests to aid in rational prescription as long as the sampling was correct, which was reported to not always be the case due to time pressure:


*“Often the material is inadequate. It is an extreme pressure in the emergency ward sometimes, and you choose the easiest solution, which gives non-representative results”*
(R6, 4 years).

The physicians experienced inadequate material as a lost opportunity to de-escalate to narrow-spectrum antibiotics.

“Collegial consulting” is bipartite; on one side, consultations with ID specialists were considered an important measure to facilitate more rational prescription by providing support in decision-making:


*“What works well is to seek collegial support when you are in doubt, call an infectious disease specialist”*
(R8, 23 years).

One the other side, the ID specialists highlighted that their antibiotic prescription advice was sometimes suboptimal due to limited time to engage, potentially hampering its effect on antibiotic prescription:


*“(…) If we had time for consultation, it would be fine, but when it (the consultation phone) buzzes on top of everything else you have to do, then you don’t have time (…), and the advice might be suboptimal”*
(R7, 13 years).

### 2.4. Theme (iv) Benevolence

The theme “Benevolence” consists of the subthemes “the patient’s wellbeing” and “treatment expectations”. This theme addresses the physicians’ benevolent efforts; they described a desire to improve their patients’ health and meet the patients’ or their relatives’ wishes, which often would include the prescription of antibiotics. Simultaneously, they expressed a desire to limit unnecessary use of antibiotics and AMR.

“The patient’s wellbeing” refers to the physicians’ prioritization of the present patient’s potential improvement, a possible future increase in antibiotic resistance notwithstanding, as illustrated by respondent 7:


*“(…) Sometimes I think, ‘Why am I doing this?’ when I, e.g., give broad-spectrum antibiotics to an old patient who has dementia and has not improved on other antibiotics—it feels wrong, but at the same time, I feel I don’t have a choice when I stand in front of the patient”*
(R7, 13 years).

The physicians expressed an unease knowing that such decisions often could result in a more inappropriate antibiotic prescription.

The “treatment expectations” subtheme represents situations when the physicians saw it as routine and expected to prescribe antibiotics, even though they often had second thoughts whether the antibiotic prescription was correct, regarding the aspect of AMR. An example some of the respondents used was oncologic or transplant patients for whom prolonged antibiotic treatment is often routine beside chemotherapeutic regimes. The physicians reported they obviously complied in offering life-extending therapy to individual patients. Nevertheless, they questioned the price to pay in terms of antibiotic resistance development:


*“It is obvious that every patient, who can and wishes to get well, should receive all available treatment (…). For some patients, this would mean complicated antibiotic regimens to keep them alive during the treatment (e.g., transplant or cancer treatment). (…) This is worrisome due to the development of resistant bacteria that may kill other individuals”*
(R1, 12 years).

Another challenge the physicians experienced was the expectations of antibiotic treatment from the patient or the patient’s family in situations when the physicians did not intend to prescribe an antibiotic:


*“You often feel pressured; you know (when a relative asks), “Can you guarantee that my father doesn’t have this and this condition?”—in situations like that you might push the boundaries more than you intended to”*
(R4, 25 years).

## 3. Discussion

Our main finding was the physicians report of the clinical workflow pressure to commonly facilitate excess antibiotic prescription to be on the safe side. Workflow pressure was perceived to hamper (1) correct microbiological specimens and (2) the quality of advice that the ID specialists gave, potentially facilitating unnecessary antibiotic prescriptions. We also identified a need to debate ethical dilemmas of antibiotic prescribing in specific clinical encounters, particularly with terminally ill patients.

The finding of clinical workflow pressure aligns with previous studies that have identified a high workload as a contributor to inappropriate prescriptions in hospitals [10,17,18]. In contrast, in two recent systematic reviews on determinants of physicians’ antibiotic prescription decisions, time pressure was not a prominent finding [9,19]. The reason for this discrepancy may relate to the fact that the authors used frameworks for analyzing cognitive determinants of physicians’ behavior. Time pressure due to high workload is mainly an external (systemic) and less of a cognitive factor and thus may not have called attention in these studies. The dominant and uniform reporting of time pressure as a barrier in our study strongly suggests that a focus on physician’s behavior alone is ineffective, a notion which is in line with a recent publication by Broom et al. [20]. Based on analyses of qualitative studies over a decade, the authors argue for “no longer portraying AMR as the result of the actions of individual healthcare professionals but instead as a systemic problem (…).” Consequently, the ability of hospital administrators to analyze the work system organization and prioritizations within may be important for the success of ASPs.

Our second most prominent finding was the physicians’ need for clinical certainty. This finding aligns with the findings of two systematic reviews that showed that fear of patient adverse outcomes causes’ antibiotic overtreatment [9,19]. Considering the physicians’ unified need to be safe, ASPs should include measures supporting prescribing physicians’ sense of safety. Our study revealed several actions that could provide such support. The physicians called for improved microbiological testing, which could be achieved by reviewing sampling techniques and reinforce training [21]. Furthermore, experience and knowledge could be accelerated by systematizing consultation with ID physicians and implementing audits and feedback on antibiotic prescription, both of which are interventions that have proven effective in ASPs [13]. In particular, attention should be paid to the less experienced physicians. They expressed the highest level of insecurity in our study and potentially need more training on when and how to use expert consultations [18].

Another finding was the physicians’ desire to do all they could for the present patient as a driver for an increased use of broad-spectrum antibiotics. However, the physicians sometimes had second thoughts about the benefit of their prescription (e.g., in pre-terminal patients), but they experienced that they did not have a choice when facing the patient. This sentiment echoes the ‘tragedy of the commons’ principle, which is known to lead to failure: If everyone thinks about themselves (misuse antibiotics now), all will suffer (increased AMR in the future) [22]. In our view, this finding highlights a need to debate when to provide and when to restrict antibiotic treatment. This topic is often too complex to be left at the discretion of the individual physician [23]. One suggestion is to develop further ethical committee services, e.g., by enabling a fast track for hospital doctors to consult with committee members. In addition, improved education on ethical topics is warranted for physicians who care for seriously ill patients, as the decision to use antibiotics is ultimately left for them [24].

In this study, we have aimed to identify factors, in a low-resistance context, that may help improve future ASPs. Factors identified in our context does not seem to diverge noteworthy from factors in other contexts [19], and thus support existing principles of ASP such as ID specialist consultation and education. However, core elements of infrastructure such as organizational structure should be reinforced [25].

Additionally, our findings of structural deficiencies may be more tangible to implement in ASPs than changes in physicians’ behavior. Additionally, structure and behavior should be seen as integral to each other [20].

For more long-term gain regarding AMR ethical questions in antibiotic prescriptions should be evaluated.

The validity of our findings was strengthened as three researchers with different backgrounds (First author; IC, last author; LPJJ, and second author; JBH) read the transcripts and agreed on the themes in the analytic process [26]. To ensure adequate report of our study we used the consolidated criteria for reporting qualitative research (COREQ) (Appendix A).

Our study also has some limitations. First, one should always be careful to generalize the results from single center studies. Still, we believe the results from our hospital in a country with a low AMR represents a valuable supplement to similar exploratory studies from locations where AMR is far more prevalent. Several of the factors we identified may be relevant to aid in developing ASPs, particularly in settings of low antibiotic resistance rates and for high-intensity antibiotic prescribers in hospitals. Secondly, the recruitment of “frequent antibiotic prescribing physicians” resulted in a high representation of ID specialists and oncologists, indicating a potential selection bias. On the other hand, this has resulted in valuable viewpoints from two perspectives, i.e., from advice-seeking prescribers and advice-giving ID specialists. Finally, the principal investigator (IC) knew most of the respondents before the interviews, which may have affected their responses and her interpretations in the analytic process. We sought to limit such influence using reflexivity throughout the process [27].

In conclusion, future ASP efforts should evaluate the system organization and prioritizations to address and manage potential time-pressure issues. To limit the use of broad-spectrum antibiotics, improving microbiology testing and the routines for consultations with infectious disease specialists seems beneficial. We also identified a need among the prescribing physicians for a debate on ethical antibiotic questions.

## 4. Materials and Methods

### 4.1. Design and Setting

This qualitative study, using semi-structured interviews, was conducted at Østfold Hospital Trust, a 380-bed, secondary, acute care hospital with a catchment area of 320,000 inhabitants. From 2016 all Norwegian hospitals began establishing ASPs and from 2017 and at the time of the study, the study hospital had an antibiotic stewardship team consisting of an infectious disease (ID) specialist, a clinical pharmacist, and a nurse trained in infectious diseases. This team mainly conducted surveillance of antibiotic use, revised antibiotic guidelines, and conducted education promoting rational antibiotic use. The national antibiotic guidelines were made available electronically and distributed as a pocket booklet. The recommendations are based on treatment tradition and national antibiotic resistance patterns. Narrow-spectrum antibiotics are recommended in several empirical regimens, e.g., benzylpenicillin is the first choice for community-acquired pneumonia, and benzylpenicillin or ampicillin combined with gentamicin is recommended for sepsis [28].

In the study hospital, similar to most Norwegian secondary acute care hospitals, the physicians consult ID specialists by phone whenever they have complicated questions regarding infections, and antibiotic treatment.

### 4.2. Recruitment

To participate in the study, we sought respondents that could give rich information on the factors affecting physicians’ antibiotic prescription. We purposively invited physicians that were prescribing antibiotics on a daily basis. Relevant respondents were chosen from lists over physicians in departments with high antibiotic use as documented by routine pharmacy data statistics and point-prevalence surveys of antibiotic prescriptions [29]. We made sure to include physicians representing different experience levels. A total of 17 physicians were invited by email and provided with information about the study, including the need to reserve 60–90 min of interview time. Of the 17 physicians invited to participate in the study, two did not have time to participate, and one did not respond, giving 14 potential respondents. Following the 12th interview, we identified no new themes. However, to ensure data saturation, we interviewed the two remaining physicians. The mean interview time was 52 min (range 23–74).

### 4.3. Interview Guide and Interviews

We developed an interview guide based on recommendations by Kallio et al. [30], and searched and used existing literature [31,32]. We designed the questions to have the respondents recall clinical situations in which they had prescribed antibiotics to obtain answers that reflected their everyday practice (Table 3). We conducted three pilot interviews with eligible candidates recruited through convenience sampling to optimize the final interview guide and technique. The data from the pilot interviews were not included in the data analysis.

**Table 3 antibiotics-11-00098-t003:** Summarized interview guide (full version in Appendix A).

1	What are your thoughts about rational antibiotic prescription?
2	What are your thoughts about antimicrobial resistance?
3	How would you describe the antibiotic prescription in this hospital, from your perspective?
4	What influences you when you prescribe antibiotics?
5	Can you please tell me about a situation where you had to decide when to start, not start, or stop antibiotics that you remember in particular?
6	Do you have any final comments on rational antibiotic prescription?

The interviews were conducted in a quiet room in the study hospital from November 2018 to February 2019. The objectives of the study were presented both orally and on a written consent form. All interviews were performed by IC, a female MD and Ph.D. student trained in qualitative methods and with clinical experience in surgery and oncology. The pilot interviews were conducted together with LPJJ, a professor of health science with prior extensive experience with qualitative research, to observe and give feedback. Since interviews were performed at the same hospital as IC was employed, both interviewer and interviewees were known to each other, either in person or as colleagues. Consequently, measures were taken to minimize the risk of bias (please see data analysis).

### 4.4. Data Analysis

The interviews were audio-recorded, transcribed verbatim, and de-identified (I.C.). In the transcripts I.C. made remarks about the physicians’ body language, pauses in speech and other non-verbal communication, to provide more context to the transcripts. To minimize the risk of bias L.-P.J.-J. and J.B.H. consecutively reviewed and gave feedback on the interview transcriptions. The respondents were not automatically offered return of transcripts, unless they inquired it. One respondent did so and read and approved of the content. I.C. and L.-P.J.-J. coded the transcripts and themes derived from the data following the recommendations of Braun and Clarke [33]. NVIVO software assisted in the coding process. I.C. scanned the transcripts for illustrative quotes. Throughout this process, the possible influence of our positions and presumptions was scrutinized and documented in a project log (confer reflexivity) [27].

## Figures and Tables

**Figure 1 antibiotics-11-00098-f001:**
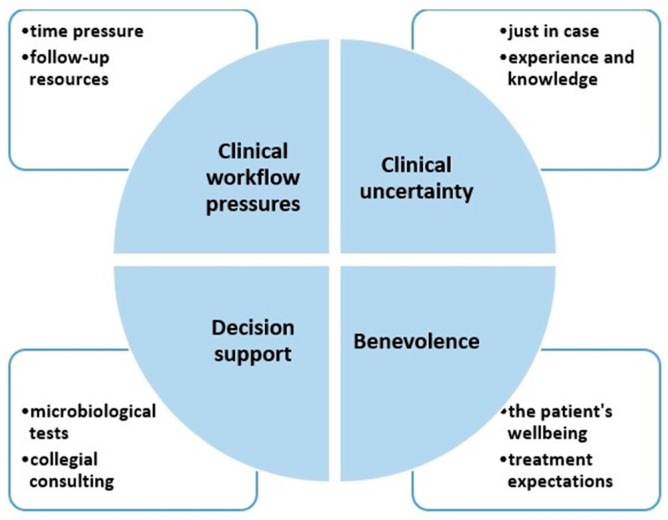
Visualization of the main themes (in the circle) with the consisting subthemes (squares).

**Table 1 antibiotics-11-00098-t001:** Characteristics of the study participants (n = 14).

**Age**	**(Years)**
Median (IQR)	36.5 (14.5)
Range	29–66
**Working Experience as a Physician (Years)**	**(n)**
<5 years	4
5–10 years	3
10–20 years	4
>20 years	3
**Gender**	**(n)**
Female	9
Male	5

IQR: Interquartile range.

**Table 2 antibiotics-11-00098-t002:** Themes, subthemes, and illustrative quotes representing the determinants of antibiotic prescription.

Theme	Subtheme	Quotes (Respondent Number, Years of Experience)
Clinical workflow pressures	Time pressure	There is too much pressure, no time to just observe the patient (…) so most doctors will give what they know works, which means broad-spectrum antibiotics, and the problem is that it works! (…) (R4, 25 years) For (patients with) chronic obstructive pulmonary disease patients and patients like that, it is difficult to know if they have an infection or not, and I feel it is more efficient just to put them on antibiotics (R14, 1 years)
	Follow-up resources	If you know the patient gets followed up, and clinical deterioration would easily get picked up, you would probably be more comfortable starting penicillin (instead of more broad-spectrum antibiotics), as opposed to a patient who gets left alone in a corridor, which is often the case (R9, 8 years) It is not the nurses’ fault, but it often happens that they say no if I ask for close follow-up (of an ill patient) during night shifts. They tell me they don’t have the capacity and no one to step in (R13, 4 years)
Clinical uncertainty	Just in case	9/10 doctors would, in cases of insecurity, rather ensure themselves (and give broad-spectrum antibiotics) (R9, 8 years) (Physicians continue antibiotics because) (…) it is the fear of doing wrong, of being appealed, both legally but also from colleagues. (R8, 23 years)
	Knowledge and experience	It is an intuition you get with clinical experience that allowed us to trust the clinical picture and withhold antibiotics (R6, 4 years) I got this idea that I wish to use narrower antibiotics, but then I don’t know enough to do so (R5, 4 years)
Decision support	Microbiological tests	The most important thing is to have a resistance pattern, then I can feel safe that I use the right antibiotic (and don’t need to give more broad-spectrum antibiotics) (R5, 4 years) We need to become better at sputum tests (…), often the patient has had a lot of productive cough and no collected sputum (…), which is a shame, because if you find the right microbe to target, then we will actually succeed (with narrow-spectrum antibiotics) (R11, 22 years)
	Collegial consulting	We often call up infectious medicine and ask for help (…) if we are in doubt. (R14, 1 years) I almost never make difficult decisions alone; I just talk to a colleague (R8, 23 years) Unfortunately, the reality is that you usually have several tasks in addition to giving consultations, and that certainly affects the quality of the advice we give (ID specialist, R4, 25 years) If you don’t have time, then the advice (on antibiotics) you give may be bad and sometimes even dangerous (ID specialist, R2, 18 years)
Benevolence	The patient’s wellbeing	If you’ve got an, e.g., immunosuppressed patient who is ill, then we treat with what is available in antibiotics now without worrying about the problem of resistance, sure we do (R10, 16 years) It is an analogy to peeing in one’s pants; it is solving the problem now (by prescribing antibiotics), everyone know it will be uncomfortable later (…), but that is a problem for tomorrow (…) (R2, 18 years)
	Treatment expectations	In cancer and transplant patients, we often choose more broad-spectrum antibiotics, as they are immunocompromised, though you know broader spectrum means more resistance (…), and that is problematic, because patients who could have been cured from cancer could later die due the complications of the treatment (multi-resistant bacteria) (R10, 8 years) When, e.g., you are 90 years old and sick, it is probably okay to die of pneumonia. We try to tell relatives that, but they have a high level of expectations and will say, “But pneumonia is treatable”, and then we start treatment anyways, and then it doesn’t work, and we often try a more broad-spectrum antibiotic, and it doesn’t work either, and that’s a vicious circle (R4, 25 years). The family was literally praying to me, ‘She must live, what can be done?’ and then we continued to treat (…), but it won’t be any better (R5, 5 years)

## Data Availability

Extended data are available upon request to the corresponding author (I.C.).

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
