# Peer review of "Factors Affecting Antibiotic Prescription among Hospital Physicians in a Low-Antimicrobial-Resistance Country: A Qualitative Study"

_antibiotics, 2022, doi:10.3390/antibiotics11010098_

Round 1

Reviewer 1 Report

The article presents a particular way of approaching the issue of antibiotic resistance, from the point of view of a hospital unit in a country where the incidence of antibiotic-resistant microbial forms is much lower than in other European countries. However, the article describes the AMR issue rather brief.

From the point of view of the approached topic, i.e. “Factors affecting the prescription of antibiotics among doctors in hospitals in a country with low antimicrobial resistance”, the level of evaluation and the evaluation methods are strictly based on interviews, without presenting concrete data or statistics, for a much longer period than the period considered in this study. This evaluation system is extremely subjective and can lead to confusion about the level of AMR in this hospital unit.

In my view this article need some major changes. Please consider the following remarks:

  1. Row 25 - In connection with patient counseling: please add some concrete examples of counseling and who performs this activity and in what context.
  2. Row 49 - In order to have a more complete picture of the differences presented in terms of AMR in Norway, compared to the rest of Europe, authors should develop this paragraph with the inclusion of more recent data in the bibliography (not only in 2014 - reference 15). What data can be provided for example from EARS-Net = European Antimicrobial Resistance Surveillance Network?
  3. Row 50-51: Authors should detail about cases of microbial resistance in this hospital. What microorganisms are involved in AMR in this hospital?
  4. Row 59: This study includes infectious disease doctors, oncologists, resident, anesthetist, gastrointestinal surgeon, gastroenterologist, pulmonologist and hematologist; without a clinical pharmacist. From my point of view it would be useful if he could have added an interview with the clinical pharmacist who is part of the medical team of AB.
  5. Row 158-159: Clinical Pharmacist Consultation: Possess extensive knowledge of the indications for a class of antibiotics, side effects and interactions in polypragmatism, frequently addressed in this study, given the frequent reference to elderly or severely ill patients (neoplasm or transplant). Authors may also include in the interview list a clinical pharmacist whose opinion may be relevant to their competence.
  6. Row 269: 4. Materials and Methods must be included before results.
  7. Row 283: 4.2. Recruitment- please present the inclusion and exclusion criteria of the study.
  8. Row 306-307: Please detail the limitations of the study.
  9. Line 309: Chapter Data Analysis:

Apart from body language, pauses in speech and other non-verbal communications, as outlined in the final decision of therapeutic approach and what place the antibiogram and the contribution of specialists (microbiologist or infection specialist, clinical pharmacist) occupy, some concrete examples of collaboration that have led to optimal therapeutic conduct are useful.

  1. Line 312: Please explain the abbreviations IC, LPJJ si JNH.
  2. The authors should include a chapter of conclusions at the end of the article.

Author Response

please see the point-by-point response in the attachment

Reviewer 2 Report

The authors report on a qualitative study on Factors affecting antibiotic prescription among hospital physicians in a low antimicrobial resistance country. Although the concept of the study is not new, the results would be useful to guide further educational activities.

Suggestions for improvement:

Introduction

Aims of the research study were clear, however, study relevance and why it was thought to be important needs more details. i.e. (lines 39-40) how would understanding and considering the factors influencing physicians’ prescribing behavior affect ASPs? this was discussed briefly in the abstract but not in the introduction section.

Methods

Authors of the study have explained how physicians were selected and why they were the most appropriate to participate in the study. However, there are no provided information about how many physicians were invited to participate in the study and why some physicians chose not to take part?

Results

Was there any contradictory data? were these considered?

Author Response

REVIEWER 2         

Thank you for your time taken and for insightful comments, we believe our manuscript has improved based on your feedback. Our replies are below; please let us know if anything is unclear. Changes in the manuscript are marked with a comment (Reviewer 2, remark #)

  1. Introduction

Aims of the research study were clear, however, study relevance and why it was thought to be important needs more details. i.e. (lines 39-40) how would understanding and considering the factors influencing physicians’ prescribing behavior affect ASPs? this was discussed briefly in the abstract but not in the introduction section.

Response: Thank you, we have now sought to better argument for the study importance, row 39-41, and 46-48.

The references 7 and 8 discusses this, often neglected, aspect of ASPs in more detail.

For instance:

In reference [7] the authors state: “(…) an understanding of the key drivers of current professional antibiotic use in hospital patients is crucial so that theories on behavioural change can be selected to generate ideas for the planning of effective interventions.”

And in reference [8] the authors accentuate qualitative studies as crucial to ensure robust interventions.

  1. Methods

Authors of the study have explained how physicians were selected and why they were the most appropriate to participate in the study. However, there are no provided information about how many physicians were invited to participate in the study and why some physicians chose not to take part?

Response: We had initially written this information under “Results”, but have now moved this information under “Methods”, row 305-310. We think it is probably given a better comprehensiveness; thank you for making us attentive on this.

  1. Results

Was there any contradictory data? were these considered?

Response: This is an interesting question. The authors who took part in the analysis (IC and LPJJ) discussed any contradictory findings during the analytic process. Noteworthy, the main results were accurate (or not mentioned) by the respondents.

There was, however, one contradictory finding. One respondent mentioned blood tests (CRP and leucocytes) as most important when making antibiotic choices. All the other physicians pointed to the clinical picture, microbiological tests, collegial consulting, or other factors as more important in the decision-making. Noteworthy, the diverging respondent, who thought blood tests were most important, had less experience (under five years). However, this data did not “count as a theme,” according to thematic analysis by Braun and Clarke, which we used.

Reviewer 3 Report

Dear Authors

I have read with interest your paper entitled "Factors affecting antibiotic prescription among hospital physicians in a low antimicrobial resistance country: a qualitative study". I think the results you provided underlined once more how inappropriate antibiotic prescription is a "systemic problem", not an individual one. These results provide evidence that (no ID) physicians should receive many more tools and training to feel more comfortable with antibiotic prescription/withdrawal.

Author Response

REVIEWER 3

Dear Authors

I have read with interest your paper entitled "Factors affecting antibiotic prescription among hospital physicians in a low antimicrobial resistance country: a qualitative study". I think the results you provided underlined once more how inappropriate antibiotic prescription is a "systemic problem", not an individual one. These results provide evidence that (no ID) physicians should receive many more tools and training to feel more comfortable with antibiotic prescription/withdrawal.

Response:

Dear reviewer 3, thank you for your comment, we agree and for antibiotic stewardship purpose; such tools must be continuously adjusted to fit the right context.

Round 2

Reviewer 1 Report

Dear Authors,

The Manuscript ID antibiotics-1539046 was improved based on my observations and became suitable for publication in Antibiotics journal.

Best regards